# Validation of GPM Rainfall and Drop Size Distribution Products through Disdrometers in Italy

Elisa Adirosi [1,*], Mario Montopoli [1,2], Alessandro Bracci [1,3], Federico Porcù [3], Vincenzo Capozzi [4], Clizia Annella [4], Giorgio Budillon [4], Edoardo Bucchignani [5,6], Alessandra Lucia Zollo [5,6], Orietta Cazzuli [7], Giulio Camisani [7], Renzo Bechini [8], Roberto Cremonini [8], Andrea Antonini [9], Alberto Ortolani [9,10] and Luca Baldini [1]

1   National Research Council of Italy, Institute of Atmospheric Sciences and Climate (CNR-ISAC), 00133 Rome, Italy; m.montopoli@isac.cnr.it (M.M.); alessandro.bracci5@unibo.it (A.B.); l.baldini@isac.cnr.it (L.B.)
2   Center of Excellence for Telesensing of Environment and Model Prediction of Severe Events, University of L'Aquila, 76100 L'Aquila, Italy
3   Department of Physics and Astronomy "Augusto Righi", University of Bologna, 40126 Bologna, Italy; federico.porcu@unibo.it
4   Department of Science and Technology, University of Naples "Parthenope", 80143 Naples, Italy; vincenzo.capozzi@uniparthenope.it (V.C.); clizia.annella@collaboratore.uniparthenope.it (C.A.); giorgio.budillon@uniparthenope.it (G.B.)
5   Meteorology Lab, Centro Italiano Ricerche Aerospaziali (CIRA), 81043 Capua, Italy; e.bucchignani@cira.it (E.B.); a.zollo@cira.it (A.L.Z.)
6   REMHI Division, Fondazione Centro Euro-Mediterraneo sui Cambiamenti Climatici, 81100 Caserta, Italy
7   Regional Agency for the Protection of the Environment of Lombardia (ARPA Lombardia), 20124 Milano, Italy; o.cazzuli@arpalombardia.it (O.C.); g.camisani@arpalombardia.it (G.C.)
8   Regional Agency for the Protection of the Environment of Piemonte (ARPA Piemonte), 10135 Torino, Italy; renzo.bechini@arpa.piemonte.it (R.B.); roberto.cremonini@arpa.piemonte.it (R.C.)
9   Consortium Laboratory of Environmental Monitoring and Modelling for the Sustainable Development (Consorzio LaMMA), Sesto Fiorentino, 50019 Florence, Italy; antonini@lamma.toscana.it (A.A.); ortolani@lamma.toscana.it (A.O.)
10  National Research Council of Italy, Institute for the Bioeconomy (CNR-IBE), Sesto Fiorentino, 50019 Florence, Italy
*   Correspondence: elisa.adirosi@artov.isac.cnr.it

**Abstract:** The high relevance of satellites for collecting information regarding precipitation at global scale implies the need of a continuous validation of satellite products to ensure good data quality over time and to provide feedback for updating and improving retrieval algorithms. However, validating satellite products using measurements collected by sensors at ground is still a challenging task. To date, the Dual-frequency Precipitation Radar (DPR) aboard the Core Satellite of the Global Precipitation Measurement (GPM) mission is the only active sensor able to provide, at global scale, vertical profiles of rainfall rate, radar reflectivity, and Drop Size Distribution (DSD) parameters from space. In this study, we compare near surface GPM retrievals with long time series of measurements collected by seven laser disdrometers in Italy since the launch of the GPM mission. The comparison shows limited differences in the performances of the different GPM algorithms, be they dual- or single-frequency, although in most cases, the dual-frequency algorithms present the better performances. Furthermore, the agreement between satellite and ground-based estimates depends on the considered precipitation variable. The agreement is very promising for rain rate, reflectivity factor, and the mass-weighted mean diameter ($D_m$), while the satellite retrievals need to be improved for the normalized gamma DSD intercept parameter ($N_w$).

**Keywords:** global precipitation measurement mission; dual-frequency precipitation radar; ground validation; disdrometer; Italy

## 1. Introduction

Satellite data are crucial to detect, measure, and monitor the precipitation amount and its characteristics at global scale. In fact, ground-based operational precipitation measuring instruments, such as weather radars or raingauges, provide useful and accurate data, but their global distribution is inhomogeneous, with some regions showing a relatively dense coverage while others having few or no devices at all (see for example http://wrd.mgm.gov.tr for the distribution of operational weather radars over the Earth and [1] for the global distribution of raingauges). In addition, ground-based instruments rarely provide measurements over the oceans. Among space-borne sensors, radar is the only one able to provide measurements resolved along the vertical, but it is also supposed to provide more accurate and highly spatially resolved rainfall rate retrieval with respect to radiometers. The Tropical Rainfall Measuring Mission (TRMM) of the National Aeronautics and Space Administration (NASA) and Japan Aerospace eXploration Agency (JAXA), launched in 1997, was the first one using radar (at Ku-band) dedicated to measuring precipitation at latitudes between 35°S and 35°N [2]. Based on the success of TRMM, at the end of February 2014, the same agencies launched the Core Satellite (CO) of the Global Precipitation Measurement (GPM) mission [3], with the aim of providing more accurate precipitation estimates for higher latitudes, ranging from 65°S to 65°N. To this purpose, a dual-frequency precipitation radar (DPR) using two frequencies at the Ku- and Ka-bands (13.6 and 35.5 GHz, respectively) was adopted with the aim of obtaining more accurate retrieval based on dual-frequency processing. The set of Level 2 products obtained from DPR measurements includes vertical profiles of the attenuation-corrected radar reflectivity ($Z$), the precipitation rate ($R$) and two of the three parameters defining the normalized gamma shaped (Equation (1)) Drop Size Distribution (DSD), namely the mass weighted mean drop diameter, $D_m$, used to describe the prevailing diameter of drops and the normalized intercept parameter $N_w$, i.e., the intercept parameter of the gamma DSD normalized with respect to the liquid water content $N_w$.

Estimates are obtained with three different algorithms, one of which takes advantage of the dual-frequency capabilities of DPR.

In general, geophysical estimates obtained from satellite measurements need to be properly validated with independent measurements. To this purpose, the GPM mission developed an extensive Ground Validation (GV) program focused around three major elements [4]: physical validation, statistical validation, and products assessment for hydro-meteorological applications. The physical validation is usually accomplished through dedicated field campaigns and physical processes studies. It uses dense clusters of purposely deployed meteorological devices along with instrumented flights (in pre- and post-launch era) for providing feedback on the assumptions used to develop retrieval algorithms (see https://gpm.nasa.gov/science/ground-validation/field-campaigns for the numerous field campaigns conducted in different regions of the globe, accessed date 1 February 2021). In the statistical validation, calibrated ground observations from operational national networks or research instruments in different regions of the world provide independent reference measurements to evaluate the performance of retrieval algorithms, after the launch, over long periods. Since differences in the performance of retrieval algorithms could depend on the regional characteristics of precipitation regimes, validation studies need to be carried out in different climatic regions of the world. Finally, the third element is the assessment of the utility of satellite precipitation products for the purposes of hydro- meteorological applications (an example focused on Italy is in [5]). In particular, the overall objective is to identify the optimal space and time scales at which satellite precipitation products will be useful for water budget studies and hydrological applications.

Since the availability of GPM data, many studies have been conducted to compare and validate the available version of precipitation products with data collected by ground-based instrumentation. Considering the validation of DPR surface rainfall products, raingauges, radar networks (for which specific ground validation products are available [4]), or a

combination of both, are the most frequently used. Raingauge validation focuses on precipitation at ground. Validation of DPR products Version 5 can be found in [6], where rainfall estimates obtained from Ku radar data were compared with different raingauge networks, showing a better performance with respect to the GPM microwave radiometer estimator. Version 6 was assessed against a dense network of raingauges in Austria, focusing on four events to conclude that dual-frequency algorithms showed the best performance [7].

Radar-based validation can consider either rainfall retrievals referred to the ground or other estimates, including microphysical parameters, resampled at a common resolution with DPR. Using the latter approach, in [8], four years of GPM-DPR Version 5 products were compared with the data obtained by five NEXRAD S-band radars in the U.S., finding good agreement in terms of co-located reflectivity with a correlation up to 0.9 at Ku-band and 0.85 at Ka-band. Comparison in terms of rainfall rate showed a correlation coefficient ranging from 0.52 to 0.69 for Ku-only and dual-frequency products, respectively.

DPR products were also evaluated using Meteo Swiss C-band radars [9]. Using Version 4A products, the authors highlighted how the GPM products are affected by the type of surface (e.g., flat terrain vs. mountains). Precipitation amounts reported in the Ku-only and dual-frequency products exhibit a small negative bias ($-0.59$ dB and $-0.68$ dB) with respect to ground radar, during summer and over flat terrains, while the GPM DPR performance is worst in winter and complex terrain. Using GPM Version 5 data, [10] found that, on average, the GPM Ku-only product underestimated the precipitation amount by 3.0 dB with respect to ground-based radar and by 3.4 dB with respect to a raingauge network in complex terrain.

Overall, the combined DPR and multi-frequency microwave imager (CMB) and DPR only products underestimate surface rain rates in comparison to the UK Radarnet product [11]. At 5-km resolution, the CMB and DPR products underestimate by 6% and 19%, respectively, while at 25-km resolution, the same products underestimate by 21% and 31%, respectively. Though the underestimates become larger at 25-km resolution, the standard deviation and correlation values significantly improve (see Table 2 of [11]), meaning that most of the random errors introduced by collocation issues are eliminated at the coarser scale. They also highlighted a seasonal dependency of the error, related to the height of the $0°$ isotherm.

Recently, in [12], three years of precipitation rate obtained with the DPR-NS Version 5 were compared with precipitation products provided by the German national meteorological service (DWD) using 17 C-band operational radars over Germany. After properly matching GPM and ground-based products, a correlation coefficient of 0.61 and a Root Mean Square Error of 1.83 mm h$^{-1}$ were found. Ground radars were also used to validate snow products (Version 6) [13,14].

Concerning Italy, the Version 4 Level 2 GPM DPR products were compared with 30-min accumulated raingauge data and instantaneous radar estimates [15]. A generally better performance of the dual frequency products with respect to the single frequency ones was found, and it was confirmed that GPM DPR estimates obtained during the warmer months are in better agreement with ground-based data.

Regarding the comparison of satellite-based and ground-based DSD parameters, in [16], DPR DSD retrievals (i.e., Version 06A of the 2ADPR algorithm) were assessed using DSD parameters obtained from more than 100 ground-based radars of the GPM Validation Network (VN). A good agreement in terms of $D_m$, in particular, for stratiform rain (Mean Absolute Error, MAE, equal to 0.22 mm) was found. In convective precipitation, the overestimation of the GPM $D_m$ is higher (MAE = 0.48), in particular, for $D_m > 2.5$ mm. The agreement in terms of $\log(N_w)$ leads to MAE = 0.32 and MAE = 0.55 for stratiform and convective rain, respectively. Investigating the $D_m$ retrieval accuracy through a comparative study of C-band ground radars and GPM products (Version 5) over Italy, a good agreement between the two estimates was found, pointed out by an absolute bias generally lower than 0.5 mm [17].

It should be noted that the version of GPM products was specified for all the studies mentioned above, because different implementations of the algorithms determine differences in products and in the relative performances.

To the best of our knowledge, very few studies have used disdrometers to validate DPR retrievals at the surface level, although disdrometers of different types, able to provide a quite direct estimation of DSD, have played an essential role in the GPM GV field campaigns, having supported the initial development of parameterizations that are used in the GPM retrieval algorithms [18]. Their accuracy has been frequently investigated in such campaigns through intercomparison experiments with different types of disdrometers [19]. However, disdrometers are still considered research instruments and, not very often, they are supported for continuous operations. Moreover, disdrometers are typically sparse, not networked and the number of disdrometers vs. satellite overpasses in coincidence with precipitation could be very scarce. One example in [20], referred likely to Version 3, compares the properties of DSDs collected by a disdrometer with those of DPR products collected over an area spanning 1° in latitude and 2° in longitude. A more recent study in [21], using Version 6 products, directly compares DPR-based DSD parameters with measurements obtained by two disdrometers at ground in the Jianghuai region in China, in 19 events collected in 2014, finding a normalized standard error around 60% between GPM DPR rainfall rate and disdrometer rainfall rate obtained considering a gamma DSD with constant shape parameter equal to 3. A ground-based Particle Image and Mass Measurement System (G-PIMMS) was instead used in Japan to validate the classification products of DPR [22]. Finally, an indirect validation of DPR DSD products can be found in [23], where a good agreement between the DSD types classified through satellite data and ground-based disdrometer worldwide is reported.

Table 1 summarizes the main characteristics of the references reported above.

**Table 1.** Main characteristic of the GPM validation studies in the literature.

| Reference Number | Ground Based Devices | GPM Sensor and Product Version | Area of Study | Main Variables Involved |
|---|---|---|---|---|
| [6] | raingauge network | DPR V05 | US, Austria and Arizona | liquid precipitation |
| [7] | raingauge network | DPR V06 | Austria | liquid precipitation |
| [8] | radar network (5 at S-band) | DPR V05 | US | Reflectivity and liquid precipitation |
| [9] | radar network (5 at C-band) | DPR V04 | Switzerland | liquid precipitation |
| [10] | radar and raingauge network | DPR V05 | Switzerland | liquid precipitation |
| [11] | radar network (18 at C-band) | DPR and CMB V05 | UK | liquid precipitation |
| [12] | radar network (17 at C-band) | DPR V05 | Germany | liquid precipitation |
| [13] | radars at S- and X-band | DPR V06 | US | solid precipitation |
| [14] | MRMS (Multi-Radar Multi-Sensor) | DPR and CMB V06 | US | solid precipitation |
| [15] | radar (22 at C- and X-band) and gauge network | DPR V04 | Italy | liquid precipitation |
| [16] | +100 radars of the GPM VN | DPR and CMB V06 | US | $D_m$ and $N_w$ |
| [17] | 3 C-band radars | DPR and CMB V05 | Italy | reflectivity and $D_m$ |
| [20] | Joss and Waldvogel disdrometer | DPR V03 | India | liquid precipitation, $D_m$ and $N_w$ |
| [21] | 2 OTT Parsivel disdrometers | DPR V06 | China | Reflectivity and liquid precipitation |
| [22] | G-PIMMS | DPR | Japan | precipitation classification |

This article aims to use laser disdrometers to validate GPM DPR Level 2 Version 6 products over Italy to compare the GPM-DPR rainfall and DSD parameters with the corresponding ones measured at ground during satellite overpasses. For the first time, a considerable number of disdrometers is used for validation purposes. The disdrometers are located in different sites, namely Rome and Florence in Central Italy, Montevergine Observatory and Capua in Southern Italy, and Bologna, Milan, and Turin in Northern Italy. Some of these sites have been recording data from before the launch of GPM.

The article is organized as follows: Section 2 provides information regarding the GPM products, the disdrometer data used in this study and processing details. In Section 3, all the available disdrometer data, not necessarily corresponding to the GPM overpasses, are analyzed in order to provide some useful information regarding the precipitation characteristics over Italy. In Section 4, the comparison methodology adopted in this study is described; while in Section 5 the GPM measurements obtained over the disdrometer locations during precipitation events have been compared with the ones measured at ground by disdrometers. Finally, Section 6 concludes the paper summarizing the main findings.

## 2. Satellite and Disdrometer Data

### 2.1. GPM DPR Data

GPM DPR products are available for different scan modes and for different algorithms, which can be single-frequency (SF), using data collected by one of the two radars, or dual-frequency (DF), using data collected by the radar operating at the two DPR frequencies.

The GPM DPR consists of two radars, one at 13.6 GHz (Ku-band) and the other at 35.5 GHz (Ka-Band). The Ku-band radar scan pattern has 49 footprints of about 5 km in diameter and therefore, its swath covers 245 km (Normal Scan mode, NS). The Ka-band radar performs two interleaved scans. In the first one (Matched Scan, MS), the Ka-band scan swath is 125 km wide and is centered in the Ku-band swath so that 25 Ka beams are matched to the central Ku beams. While the Ku-band radar completes the NS scan, the Ka-band radar scans in the high sensitivity scan mode (HS), so as to observe almost the same area of the matched scan (inner swath). For NS and MS modes, the range resolution is 250 m, while for the HS mode, the range resolution is 500 m for improving the sensitivity. By the way, L2 products are provided with nominal resolutions of 125 m for NS and MS modes and of 250 m for the HS mode. Note that on May 21, 2018, the scan pattern of the HS mode was changed, and now the Ka-HS beams scan in the outer swath and are matched with Ku-NS beams. In this way, it will be possible to apply DF algorithms to the full Ku swath. Currently, products obtained for this swath are under verification and therefore are not used in this study. The reference source of information about the precipitation Level 2 (L2) products (version 6A) obtained from DPR measurements used in this study is [24]. All the precipitation products are obtained through the same modular procedure, but important differences in the modules characterize DF with respect to SF products. Therefore, it is worth summarizing such procedures, highlighting the differences between the products. The role of each module (identified by italic capital letters) is described in the following. The Preparation (PRE) module prepares raw Level 1 input products and external information to be used by the other modules. Using the measurement of the power received by the two radars, system parameters, orbit, and scan geometry, it computes the reflectivity factors, reduces the influence of clutter, identifies the clutter-free bin closest to the terrain, and classifies the pixels with precipitation that will undergo further processing. In addition, it provides the measurements of the normalized surface cross section (NRCS) used for the attenuation correction along with surface type classification. The vertical profile (VER) module computes the path-integrated attenuation due to non-precipitation particles using ancillary environmental data from the Japan Meteorological Agency (JMA), namely from the JMA Global Analysis (GANAL), such as pressure, temperature, water vapor, and cloud liquid water. The attenuation from ice particles is neglected at both Ka and Ku bands, while the attenuation due to cloud liquid water content is estimated [25]. The Classification (CSF) module classifies precipitation types and provides information on bright-band (for SF products) or melting layer (for DF products) through distinct algorithms that provide at least three major classes, namely stratiform, convective, and "other". Recent versions of the algorithms include other flags characterizing non-liquid precipitation (see [13,26]). The DSD (Drop Size Distributions) is an important module that sets the physical variables relative to precipitation particles, such as density, dielectric constants, falling velocity, and the relations used by the solver module (see below). Based on different CSF outputs, a

profile is subdivided through nodes that imply the use of different particle models. In general, particles are modeled as a mixture of air, water, and ice expressed with different volume ratios. The drop size distribution is assumed to follow a normalized gamma model

$$N(D) = N_w D^\mu \exp\left[-\frac{(4+\mu)\, D}{D_m}\right] \tag{1}$$

with the shape parameter $\mu$ fixed to 3 so that only the two parameters $N_w$ and $D_m$ are needed to describe the DSD. This simplifying assumption was found to be appropriate for dual-frequency radar retrievals that have to rely on no more than two measurements, namely the reflectivity factors measured by the radars using Ku- and Ka-frequency bands [27]. It should be noted that the gamma DSD assumption, even with three parameters, is common but not always the most appropriate [28].

Important relations for estimating precipitation parameters are set specifically for both frequencies, between $D_m$ (properly defined also for melted and solid particles) and other variables, such as $k/N_w$, where $k$ is the specific attenuation in dB km$^{-1}$, $Z_e/N_w$, where $Z_e$ is the effective reflectivity factor, and finally, a relation between $D_m$ and $R/N_w$, with R being the precipitation rate in mm h$^{-1}$ for temperature ranging between $-50\,°C$ and $50\,°C$ with $1\,°C$ interval. The module SRT (Surface Reference Technique) computes the path-integrated attenuation (PIA) due to propagation through precipitation using the radar returns from the surface: it is assumed that the difference between the measurements of the NRCS in rain and clear air condition provides an estimate of the PIA [29]. The method is applied independently to each frequency. Taking advantage of the correlation in the NRCS at the two frequencies, a dual-frequency-derived path attenuation at the Ku- and Ka-band is also generated in the inner swath. Slightly different variations of the technique are run and a combination of them provides the final PIA estimate [27]. From Version 6, in addition to the SRT estimates, also a hybrid estimation, based on both SRT (single and dual frequency) and the Hitschfeld–Bordan method (HB) [30], is obtained. Finally, the Solver (SLV) module obtains DSD parameters and precipitation rates at each range bin with

$$R = \varepsilon^\tau \alpha D_m{}^\beta \tag{2}$$

where $\alpha$, $\beta$, and $\tau$ are constants equal to 0.401, 6.131, and 4.649 for stratiform rain and 1.370, 5.420, and 4.258 for convective rain, respectively. The equation includes an adjustment factor $\varepsilon$, conceived to reconcile inconsistencies between attenuation estimates obtained by the different attenuation estimation techniques [31] that takes a single value along the precipitation profile. Having assumed a gamma DSD with a fixed shape parameter, it is possible to establish a relation between R and $D_m$ for various effective reflectivities. In this way, given an effective reflectivity factor and $\varepsilon = 1$, an (R, $D_m$) pair can be obtained, and, using the tables established in the DSD module, the corresponding $N_w$ and $k$ can be obtained as well. The process starts from the top, where the measured reflectivity is supposed to be unaffected by attenuation and can be corrected iteratively using the estimated $k$. Once the procedure is applied to the entire column, a PIA profile is also obtained. The process is repeated with different $\varepsilon$ and the one minimizing the retrieved PIA at the surface level with the SRT-estimated PIA is chosen. Different minimization criteria are used for DF and SF products. Details of the procedure and implications for SF and DF algorithms are analyzed in [32].

The SF product at Ku-band is available for pixels in the whole swath of the NS mode. The SF products at Ka-band are available for pixels in the inner swath of the MS mode and in the swath of the HS mode (until 18 May 2018). Finally, the DF products are provided for pixels in the inner and outer swaths: for single-beam pixels the DF algorithm can use data in dual-frequency observations at neighboring pixels. This study uses the DF-based 2ADPR-NS, 2ADPR-MS and 2ADPR-HS products, the SF-based 2AKa-MS, 2AKa-HS and 2AKu-NS Version 6 Level 2 DPR products. Note the 2ADPR-NS and the 2ADPR-MS rain rate and DSD products coincide in the inner swath. Among all the different output vari-

ables available in the 2ADPR and 2AKu/2AKa products, for the comparative analysis, we selected: precipRateNearSurface (RNS), namely, the precipitation rate (mm h$^{-1}$) estimated at the clutter free bin nearest to the surface (binClutterFreeBottom, CFB), the zFactorCorrectedNearSurface (ZNS), namely the reflectivity factor with attenuation correction (in dBZ) at the CFB; and the paramDSD, namely the normalized gamma DSD parameters $N_w$ (in mm$^{-1}$ m$^{-3}$) and $D_m$ (in mm), evaluated at the CFB (for further information on these products, see [24]).

The mean height of the CFB above the considered disdrometers ranges between 0.60 km and 1.48 km, depending on the orography around the disdrometer and the considered GPM product. In fact, the height of the CFB depends on the local zenith angle that increases as a function of the distance from the nadir. For the considered locations, at the edges of outer swath of NS, the difference of the CFB height with respect to that at nadir can exceed 1 km, while it is more limited in the inner swath. GPM algorithms obtain estimates at a bin corresponding to surface level (binRealSurface) through extrapolation. In order to investigate the effectiveness of such an extrapolation, we compared the rainfall rate, $D_m$ and $N_w$ obtained at the CFB with the ones extrapolated at the binRealSurface level for the locations of the considered disdrometers. The comparison shows very small differences between the GPM estimates at the CFB and those referred at ground, with Normalized Mean Absolute Error (NMAE) less than 1% for DSD parameters and less than 9% for the rainfall rate. Therefore, in this article, we will consider only the GPM products at the CFB. Please note that we did not perform the latter comparison for the reflectivity factor because, below the CFB, it is assumed to be constant by the GPM algorithm, and the same value at the CFB is used in the cluttered bins [24].

Furthermore, to select only rainy overpasses, the height of the CFB is compared with that of the bottom of the bright band (namely the binBBBottom, BBB) for DPR-NS, Ka-HS, Ka-MS, and Ku-NS products and with the height of the melting layer bin (namely binDFRmMLBottom, MLB) for DPR-MS and DPR-HS products. When the BBB or MLB are not available, the forecasted height of the 0 °C isotherm is considered.

### 2.2. Disdrometer Data

Although a disdrometer network is not present in Italy, in recent years, several Italian Institutions decided to purchase and run a disdrometer. In this study, thanks to a spontaneous collaboration among different institutions, we tried to collect and use disdrometer data available in different Italian regions. For GPM DPR validation purposes, the following datasets have been considered:

- Rome: Thies Clima laser disdrometer (TC) installed during 2012 on the roof of the building of the Institute of Atmospheric Sciences and Climate (ISAC) of the National Research Council (CNR) of Italy in Rome (hereinafter TC-RM). The owner of the device is the Regional Agency for the Protection of the Environment of Piemonte (ARPA Piemonte).
- Milan: TC installed on the roof of the main building of the Regional Agency for the Protection of the Environment of Lombardia (ARPA Lombardia) in Milan (hereinafter TC-MI). The owner of the device is ARPA Piemonte.
- Turin: TC installed during 2006 in Turin (hereinafter TC-TO). The owner of the device is ARPA Piemonte. This is the older version of the TC disdrometer.
- Montevergine Observatory: TC installed on the roof of the Montevergine's monastery. It is part of the Montevergine meteorological observatory, located in the Southern Apennines, about 45 km east of Naples urban area (hereinafter TC-NA). The owner of the device is the University Parthenope [33].
- Florence: OTT Parsivel2 disdrometer (P2) installed on the roof of the Institute of BioEconomy (IBE) of CNR in Florence (hereinafter P2-FI). The owner of the device is ISAC-CNR.

- Bologna: P2 installed on the rooftop of the Department of Physics and Astronomy "Augusto Righi" of the University of Bologna (hereinafter P2-BO). The owner of the device is the University of Bologna.
- Capua: P2 installed on the roof of the Italian Aerospace Research Centre (CIRA) in Capua (CE) (hereinafter P2-CE). The owner is CIRA.

To summarize, the locations of the disdrometers used in this study are shown in Figure 1, while Table 2 reports the coordinates of all the devices and the time period considered in this study. GPM uses the World Geodetic System-84 (WGS-84) as reference ellipsoid and the disdrometer coordinates are referred to the same system. With the exception of the TC-NA, the disdrometers are located in a relatively flat terrain, at least within a 5-km distance. For these disdrometers, the height of the CFB depends mostly on the considered GPM mode. On average, the height of the CFB above the disdrometer is 0.9 km for Ka-MS, while for the other modes, it is around 1.35 km. For TC-NA, located at 1280 m height, for Ka-MS, the CFB is just 0.6 km above the disdrometer, while for Ka-HS, it is 0.85 km and for the other modes is around 1 km.

Following the Köppen–Geiger climate classification [34], all the disdrometers are located in group C (temperate climate); however, the TC-MI, TC-TO and P2-BO fall into the Csc (Mediterranean cold summer climates) area while the others in the Csa (Mediterranean hot summer climates) area. Furthermore, the TC-NA is the only device located in a mountain environment. Data collected by two different types of disdrometer, i.e., TC and P2, have been used in this study: most of the data were collected by TC (namely 79.5%) and 20.5% by P2. Both disdrometers have their own strengths and shortcomings in the measurements of the hydrometeor spectra that influence the estimates obtained from them. Some information regarding the impact of these disdrometers on the estimation of DSD and rainfall parameters, and weather radar algorithms as well, evaluated from co-located measurements processed like in this study, can be found in [35]. Considering the results of that study, all the disdrometer-estimated DSDs have been considered as "true", and no correction has been applied to account for the differences in the type of the disdrometer used for the data collection.

**Table 2.** Information regarding the different devices used in this study. Coordinates are referred to the WGS-84 reference ellipsoid.

| Device | Label | Location | Latitude | Longitude | Height ASL (m) | Time Period Considered |
|--------|-------|----------|----------|-----------|----------------|------------------------|
| TC | TC-RM | Rome | 41.8425 | 12.6464 | 102 | Feb. 2014–Oct. 2020 |
| TC | TC-MI | Milan | 45.4904 | 9.1947 | 150 | Apr. 2014–Apr. 2015 Jan. 2018–Oct. 2020 |
| TC | TC-TO | Turin | 45.0294 | 7.6549 | 250 | Feb. 2014–Oct. 2020 |
| TC | TC-NA | Montevergine's Observatory | 40.9365 | 14.7291 | 1280 | Dec. 2018–Oct. 2020 |
| P2 | P2-FI | Florence | 43.7977 | 11.1918 | 40 | Dec. 2018–Oct. 2020 |
| P2 | P2-BO | Bologna | 44.4993 | 11.3538 | 65 | Dec. 2018–Oct. 2020 |
| P2 | P2-CE | Capua | 41.1192 | 14.1721 | 70 | Jul. 2015–Oct. 2020 |

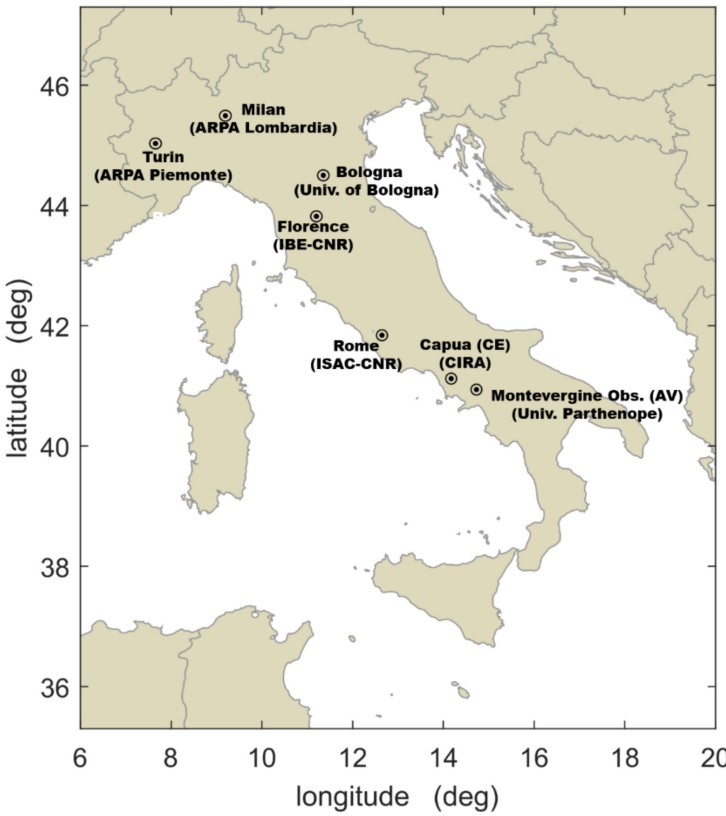

**Figure 1.** Locations of the disdrometers used in this study.

All the devices provided data each minute, and all the datasets have been pre-processed following the same procedure. Firstly, in order to filter out the spurious drops due to wind effects, splashing or mismatching, and to eliminate non-liquid hydrometeors, a filter criterion based on the fall velocity has been applied to the raw disdrometer spectra. The adopted criterion eliminates drops with a fall velocity outside the $\pm 50\%$ of the [36] diameter-fall velocity relation. Then, only 1-min samples with at least 11 drops have been considered to compute:

$$N^{P2;TC}(D_i) = \frac{1}{A^{P2;TC}\Delta t\,\Delta D_i^{P2;TC}} \sum_{j=1}^{C_v^{P2;TC}} \frac{n_{j,i}}{v_j} \tag{3}$$

where the superscript indicates the specific instrument, N(D) is the drop size distribution $(mm^{-1}\,m^{-3})$, $\Delta t$ is the sampling time (namely 60 s), A is the instrumental measuring area $(m^2)$, v $(m\,s^{-1})$ is the fall velocity from the theoretical diameter-fall velocity relation [36], $\Delta D$ is the width of the size bin, $n_{i,j}$ is the number of drops measured in the i-th diameter class and j-th fall velocity class, and $C_v$ is the total number of fall velocity bins. The width of each diameter class is provided by the manufacturers. In order to consider only liquid precipitation samples, an air temperature criterion has been added to the fall velocity one, based on air temperature measured by weather stations close to the disdrometers, to filter out DSD samples likely contaminated by mixed or solid precipitation. Finally, for each DSD, the radar reflectivity factor at Ka- and Ku-band ($Z_{Ka}$ and $Z_{Ku}$, respectively), R, $D_m$, and $N_w$ can be computed as

$$Z_{Ka,Ku} = \frac{\lambda^4\,10^{18}}{\pi^5\,|K_w|^2} \sum_{D_{min}}^{D_{max}} \sigma_{ka,ku}(D)\,N(D)\,dD \quad \left(mm^6\,m^{-3}\right) \tag{4}$$

$$R = 6\,\pi\,10^{-4} \sum_{D_{min}}^{D_{max}} v(D)N(D)D^3\,dD \quad \left(mm\,h^{-1}\right) \tag{5}$$

$$D_m = \frac{\sum_{D_{min}}^{D_{max}} N(D)D^4 dD}{\sum_{D_{min}}^{D_{max}} N(D)D^3 dD} \quad (mm) \tag{6}$$

$$N_w = \frac{256}{\pi \, \rho_w} \frac{10^3 \, LWC}{D_m^4} \quad \left(mm^{-1}m^{-3}\right) \tag{7}$$

$$LWC = \frac{\pi \, 10^{-3}}{6} \, \rho_w \, \sum_{D_{min}}^{D_{max}} N(D)D^3 dD \quad \left(g \, m^{-3}\right) \tag{8}$$

where LWC is the liquid water content, $\lambda$ is the wavelength (in m), $K_w$ is the complex dielectric constant of water, $\rho_w$ is the density of water (1 g cm$^{-3}$) and $\sigma_{Ku,Ka}(D)$ are the backscattering radar cross section (in m$^2$) for Ku- and Ka-band of a drop of equivalent diameter D. Hydrometeor scattering properties depend on several factors, such as composition, shape, orientation and size of the scatters, and the radar wavelength. For this study, the T-matrix method ([37,38]) was applied to compute the backscattering cross section of oblate hydrometeors. To perform the electromagnetic simulation, we assumed (i) an environmental temperature of 20 °C, (ii) that the shape of the hydrometeors follows the model proposed by [39] and (iii) the distribution of the hydrometeor canting angles is modeled with a Gaussian distribution with mean 0° and standard deviation 10° [40]. Considering only samples with R > 0.1 mm h$^{-1}$, we obtained more than 580,000 usable samples.

### 3. Precipitation Characteristics from Disdrometer Data

In this section, the main characteristics of the precipitation measured by different disdrometers over Italy are illustrated and discussed. Figure 2 shows the histograms of the rainfall (namely $Z_{Ka}$, $Z_{Ku}$, R) and DSD (namely $N_w$ and $D_m$) parameters obtained from the DSD datasets collected by different disdrometers over Italy, while Table 3 shows, for each parameter, the mean, mode, and median values of every single dataset and of all datasets together. In terms of mean and median values of R, $D_m$ and $\log_{10}(N_w)$, the different datasets are comparable, with a bit more differences for $Z_{Ka}$ and $Z_{Ku}$ and larger $D_m$ for the further southern disdrometers (TC-RM, TC-NA, TC-CE). Figure 3 shows the 2D histograms between $\log_{10}(R)$ and $D_m$ obtained from disdrometer data over Italy. As explained in the previous section, in the GPM-DPR algorithm, different R-$D_m$ relations are adopted for convective and stratiform rain conditions. The mathematical forms of the latter relations can be found in [24], and, in both cases, a parameter $\varepsilon$ is present on the right side of the expression that is an adjustment factor that is set to vary between 0.2 and 5. The best $\varepsilon$ value is the one that provides the minimum differences between the calculated and estimated PIA. In Figure 3, the disdrometer $D_m$ estimates are plotted versus the disdrometer-based rainfall rate values along with the R-$D_m$ relations used in the GPM-DPR algorithm for $\varepsilon = 1$ (solid lines) and for $\varepsilon = 0.2$, and $\varepsilon = 5$ (respectively, lower and upper dashed lines), for convective (red lines) and stratiform (black lines) conditions. Most of the disdrometer measured data are between the two solid lines, in particular, for small values of $D_m$, while for $D_m > 1.5$, the disdrometer measurements tend to drift away with respect to the GPM-DPR relations.

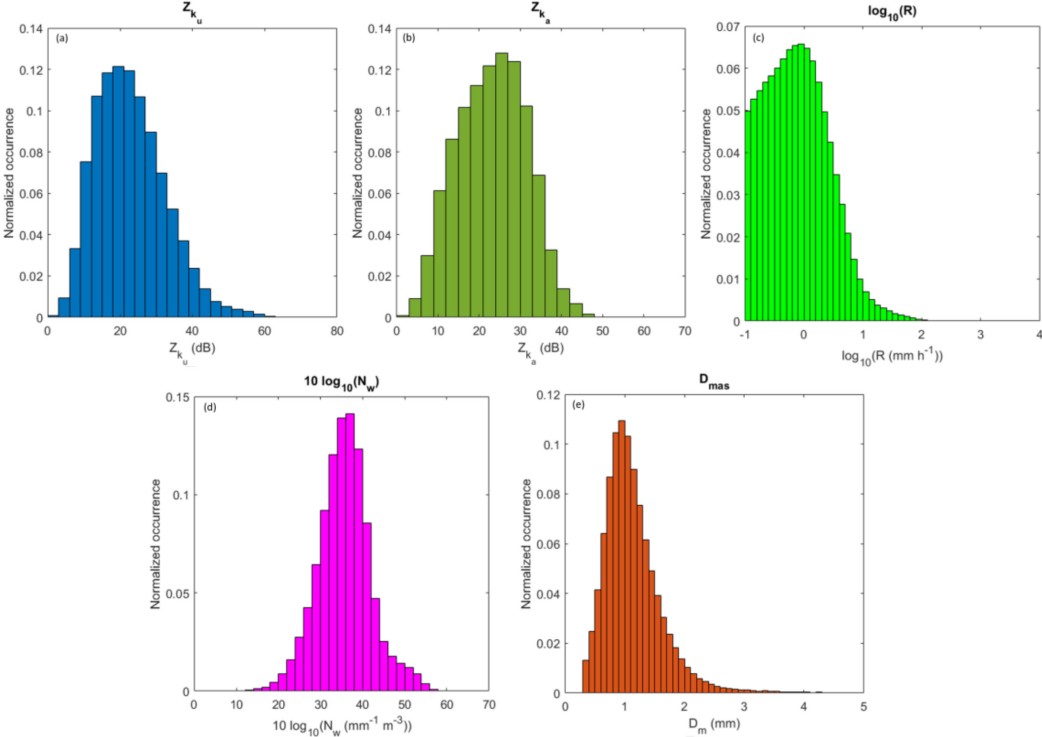

**Figure 2.** Histograms of (**a**) $Z_{Ku}$, (**b**) $Z_{Ka}$, (**c**) $\log_{10}(R)$, (**d**) $10 \log_{10}(N_w)$ and (**e**) $D_m$ as derived from all the disdrometer datasets.

**Table 3.** Mean and median values of the different rainfall and DSD parameters considered in this study for each single dataset and for all the datasets together.

|  | $Z_a$ | | $Z_{Ku}$ | | $R$ | | $10 \log_{10}(N_w)$ | | $D_m$ | |
|---|---|---|---|---|---|---|---|---|---|---|
|  | **Mean** | **Median** | **Mean** | **Median** | **Mean** | **Median** | **Mean** | **Median** | **Mean** | **Median** |
| **TC-RM** | 24.52 | 24.82 | 24.51 | 23.33 | 2.48 | 0.79 | 34.15 | 34.34 | 1.26 | 1.15 |
| **TC-MI** | 22.59 | 22.77 | 22.19 | 21.13 | 1.98 | 0.71 | 37.46 | 36.97 | 1.06 | 1.00 |
| **TC-TO** | 22.73 | 23.12 | 22.23 | 21.49 | 1.75 | 0.75 | 36.97 | 36.92 | 1.07 | 1.00 |
| **TC-NA** | 24.18 | 24.65 | 23.88 | 23.29 | 1.94 | 0.75 | 35.19 | 35.18 | 1.18 | 1.13 |
| **P2-FI** | 21.62 | 21.29 | 21.14 | 19.89 | 1.83 | 0.61 | 36.22 | 36.10 | 1.07 | 0.99 |
| **P2-BO** | 22.32 | 21.61 | 22.08 | 20.09 | 1.85 | 0.62 | 35.20 | 35.38 | 1.14 | 1.02 |
| **P2-CE** | 23.53 | 23.50 | 23.45 | 22.01 | 1.99 | 0.64 | 32.78 | 32.90 | 1.27 | 1.16 |
| **All** | 23.20 | 23.44 | 22.88 | 21.85 | 1.97 | 0.73 | 35.70 | 35.72 | 1.14 | 1.05 |

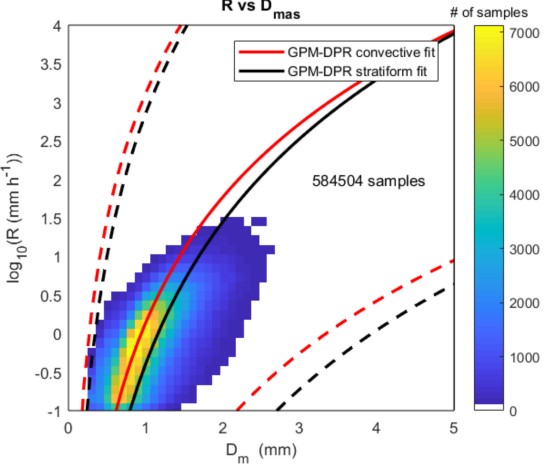

**Figure 3.** 2D histogram of $\log_{10}(R)$ vs $D_m$ obtained from disdrometers, the red (black) solid curve represents the relation used in GPM algorithm for convective (stratiform) rain in $\varepsilon = 1$, while the dashed lines are for $\varepsilon = 0.2$ (lower curves) and $\varepsilon = 5$ (upper curves).

## 4. Comparison Approach

Different strategies can be used to compare point measurements collected at ground and areal measurements collected aloft. A simple strategy is to select simultaneous measurements of the disdrometer and of the $5 \times 5$ km$^2$ DPR pixel at ground that contains the location of the disdrometer, which can be elsewhere in the pixel. The DPR estimates closest to the ground are those corresponding to the binRealSurface. Actually, as shown above, such an estimate is obtained by extrapolating at ground the estimates obtained at the height of the bin unaffected by clutter close to the surface and, since we found a negligible difference between this estimate and the extrapolation at ground, we consider the estimate at the CFB. In this point comparison, the disdrometer measurement is taken as representative of the areal estimate and, therefore, the comparison result will be affected by this source of error. The variability at small scale of the variables of interest was studied with dense networks of disdrometers during GPM-GV campaigns ([41,42]). It was found that the spatial correlation decreases with distance depending on the nature of the precipitation (stratiform/convective) and of the considered variable. Significant DPR estimates are determined at the CFB, but due to advection, precipitation sampled at the CFB can fall outside the corresponding pixel at the surface. Since it is not possible to determine exactly the DPR pixels from which the precipitation measured by the disdrometer come from, to take this into account, two strategies are pursued. First, the DPR pixels whose centers are within 5 km from the disdrometers are averaged (hereinafter referred to as mean). The second strategy considers the DPR pixel at ground containing the disdrometer and the 8 neighboring pixels (i.e., the DPR pixels in a $3 \times 3$ box around the disdrometer) and selects the DPR pixel whose reflectivity value is closest to that estimated by the disdrometer (hereinafter referred to as optimal [43] Assuming that the pixel selected with the optimal method is the one where the precipitation collected at ground by the disdrometer comes from, the optimal method should provide the best match between satellite and ground-based data. Considering the mean comparison mode, the maximum number of selectable DPR pixels is 5. The choice of such a radius considered in the mean comparison mode has been driven by the results of a sensitivity study and by physical considerations based on the possible displacement of a drop between the CFB and the ground due to horizontal wind. As an example, assuming a mean drop fall velocity of 5 m s$^{-1}$, a CFB height of 1500 m above the disdrometer and a wind speed of 10 ms$^{-1}$, a drop can be shifted horizontally by 3 km, that becomes 6 km for a wind speed of 20 ms$^{-1}$. A sensitivity study considering the effect of choosing a 5-km or 10-km radius is also performed. The maximum increment of the number of overpasses considering 10 km instead of 5 km is 8 (increment of the 11%) but, in most of the cases, the comparison performances are better when a 5-km radius is selected. Considering all the above, we decided to select the 5-km radius. Finally, only the overpasses with at least one pixel with RNS > 0.1 mm h$^{-1}$ from those selected were retained.

Considering all the disdrometer locations analyzed in this study, the number of selected overpasses for each GPM product and scan mode are shown in the first and second column of Table 4: in the first column, the DPR pixels in a 5-km radius are selected while, in the second column, the 9 pixels around the disdrometer are considered. The number of overpasses in the second column is a bit higher because the number of selected DPR pixels is higher with respect to those selected in a circle of 5 km radius centered on the disdrometer site. Again, please note that the DPR-HS and Ka-HS have a lower number of overpasses because we consider only the data collected before the change of the scan patterns occurred in May, 2018. The differences in the number of overpasses between DF and SF products for similar swath are due to differences in the SF and DF rain rate estimations algorithms that caused a different distribution of RNS values around the threshold of 0.1 mm h$^{-1}$. The difference in terms of the number of overpasses between DPR-MS and Ka-MS is relatively high and is due to the sensitivity of the Ka-band in measuring very low precipitation values. In fact, there are samples where Ka product do not report precipitation, while Ku provides

a low precipitation value (less than 0.5 mm h$^{-1}$). In all these cases, we have the DPR-MS product, but we do not have the Ka-MS product.

After selecting the GPM overpasses in rainy conditions, the disdrometer data within $\pm\Delta t$ minutes with respect to the GPM overpass time were selected and averaged. The interval $\Delta t$ was set equal to 5 min. It should be noted that the GPM data are instantaneous and referred to a footprint at ground of roughly 5 km $\times$ 5 km, while the disdrometer provides a measurement each 1 min over a surface negligible with respect to the satellite footprint, so it is essentially a point measurement. Therefore, to account for the different time and space sampling, we resorted to time averaging the disdrometer time series over a 10-min window (the minimum number of consecutive disdrometer samples to retain the average has been set to 3). A sensitivity study was performed varying the time window width from 4 min (i.e., $\Delta t = 2$) to 10 min (i.e., $\Delta t = 5$). Assuming a mean drop fall velocity of 5 m s$^{-1}$, a drop takes 6 min to fall from a height of 1800 m and a 10-minutes interval can be considered sufficient to take into account both the fall of drops to the ground and a possible non-perfect synchronization of the clocks of the disdrometer and satellite. Considering the reflectivity values, the 10-min time window for averaging seems to be a good trade-off between the number of samples considered and the performance of the comparison between satellite and ground-based data. However, it should be highlighted that the differences in terms of performance and number of samples between the different integration time windows tested are very small, suggesting that the influence of the width of the time window is limited. Considering the point comparison mode, the differences in the number of matched satellite–disdrometer samples range between 2 and 10, depending on the GPM product considered, while the differences in terms of Mean Absolute Error (MAE) are less than 0.1 mm h$^{-1}$, 0.4 dB, 0.08 mm and 0.6 log$_{10}$(mm$^{-1}$ m$^{-3}$) for R, Z, D$_m$ and N$_w$, respectively. Some information regarding the temporal variability of DSD within 10 min is provided in the next section.

Considering all the available disdrometer datasets, the number of matched disdrometer data and GPM overpasses are shown in Table 4 (third, fourth and fifth column for point, mean and optimal comparison mode, respectively). The comparison has been made in terms of Z, R, D$_m$, and N$_w$. We recall here that in the inner swath, the DPR-NS rainfall rate and DSD parameters coincide with those of DPR-MS, while the reflectivities differ because the reflectivity at Ku-band is reported in DPR-NS products and that at Ka-band is reported in the DPR-MS. The agreement between the satellite and ground data is evaluated considering the Normalized Bias (NB, in %), the NMAE (in %), the MAE (in the same unit of the considered variable), and the Pearson correlation coefficient (corr). Negative NB values indicate an underestimation of the GPM product with respect to the disdrometer measurement.

**Table 4.** For each GPM product the number of overpasses with at least one pixel with RNS > 0.1 mm h$^{-1}$, the number of matched disdrometer and DPR data for point, mean and optimal comparison mode are reported.

| GPM Product | # ovp. with Rain (Pixels within 5 km from Disdrometer) | # ovp. with Rain (9 Pixels around the Disdrometer) | # Matched Data (Point) | # Matched Data (Mean) | # Matched Data (Optimal) |
|---|---|---|---|---|---|
| DPR NS | 261 | 342 | 54 | 61 | 68 |
| DPR MS | 132 | 173 | 29 | 31 | 36 |
| DPR HS | 69 | 88 | 11 | 17 | 19 |
| Ka HS | 75 | 91 | 11 | 17 | 20 |
| Ka MS | 97 | 135 | 22 | 28 | 33 |
| Ku NS | 259 | 340 | 53 | 61 | 68 |

## 5. GPM DPR and Disdrometer Comparison

In this section, the Z, R, $D_m$, and $N_w$ obtained from satellite data are compared with those measured by optical disdrometers in the various Italian peninsula sites (see Figure 1) following the comparison approach described in the previous section.

Table 5 reports the merit factors obtained comparing satellite and disdrometer data in the point, mean and optimal comparison mode, while Figure 4 shows the scatterplots between satellite and ground-based variables. Since the optimal comparison mode generally outperforms the mean comparing mode, Figure 4 only shows the point and optimal comparison mode. In the table, the correlation values that are not statistically significant are indicated with the symbol $^{(*)}$. The significance has been tested through the $t$-test ([44]) with a significance level $\alpha = 0.05$. The latter indicates the cases where there is not a significant linear relationship between satellite and ground-based data. In the following, the performance of each single retrieved variable is discussed.

**Table 5.** Merit parameters of the comparison between GPM and disdrometer data. NMAE and NB are in %, while MAE is in the same unit of the variable, as reported in the table. Finally, corr is dimensionless. For each variable and comparison mode, the bold highlights the best score and the underlined the worst, while the symbol $^{(*)}$ indicates the non-statistically significant values.

| | | Mean | | | | Point | | | | Optimal | | | |
|---|---|---|---|---|---|---|---|---|---|---|---|---|---|
| | | NMAE (%) | MAE | NB (%) | Corr | NMAE (%) | MAE | NB (%) | Corr | NMAE (%) | MAE | NB (%) | corr |
| **R** (mm h$^{-1}$) | DPR-NS | 64.3 | 1.00 | 27.5 | **0.72** | 72.4 | 1.02 | 28.7 | 0.52 | 63.4 | 0.95 | 21.9 | 0.67 |
| | DPR-MS | 52.9 | 0.79 | 10.9 | **0.72** | 52.4 | 0.83 | **4.15** | **0.73** | 45.4 | 0.66 | 2.76 | 0.77 |
| | DPR-HS | 52.9 | 0.72 | −33.5 | 0.62 | 40.8 | 0.77 | −22.1 | 0.63 | 30.3 | 0.38 | −18.5 | **0.80** |
| | Ka-HS | **51.4** | **0.70** | −32.2 | 0.67 | 41.1 | 0.78 | −19.7 | 0.62 | 42.9 | 0.62 | −31.8 | 0.66 |
| | Ka-MS | 73.4 | 1.31 | −9.07 | 0.35 $^{(*)}$ | 66.6 | 1.38 | −16.9 | 0.39 $^{(*)}$ | 51.0 | 0.80 | −12.1 | 0.66 |
| | Ku-NS | 72.5 | 1.12 | 24.5 | 0.68 | 71.8 | 1.03 | 14.7 | 0.50 | 50.3 | 0.76 | **1.32** | 0.75 |
| **Z** (dBZ) | DPR-NS | 18.7 | 4.78 | 5.88 | 0.71 | 20.0 | 5.04 | 8.06 | 0.64 | 10.4 | 2.59 | 2.83 | 0.86 |
| | DPR-MS | **12.9** | 3.27 | 3.34 | **0.76** | 14.1 | 3.63 | 3.10 | 0.59 | 8.09 | 2.00 | 1.35 | 0.81 |
| | DPR-HS | 15.8 | 3.95 | −6.53 | 0.64 | 8.83 | 2.51 | −5.71 | **0.72** | 7.20 | 1.75 | −0.79 | **0.87** |
| | Ka-HS | 15.0 | 3.76 | −5.62 | 0.66 | 8.93 | 2.53 | −5.05 | 0.70 | 8.13 | 2.02 | −3.26 | 0.83 |
| | Ka-MS | 14.5 | 3.84 | **2.89** | 0.56 | 13.9 | 3.86 | **0.29** | 0.51 | 11.6 | 2.94 | 5.65 | 0.79 |
| | Ku-NS | 18.7 | 4.78 | 5.97 | 0.72 | 20.0 | 5.07 | 7.77 | 0.64 | 10.5 | 2.63 | 2.49 | 0.86 |
| **D$_m$** (mm) | DPR-NS | 25.1 | 0.32 | 8.51 | 0.53 | 24.6 | 0.32 | 8.15 | 0.58 | 22.9 | 0.29 | 7.29 | 0.64 |
| | DPR-MS | **22.0** | **0.29** | **0.40** | **0.64** | **21.7** | 0.29 | 1.48 | **0.67** | 26.0 | 0.34 | −1.82 | 0.21 $^{(*)}$ |
| | DPR-HS | 27.1 | 0.38 | −7.98 | 0.32 $^{(*)}$ | 28.9 | 0.47 | −12.1 | 0.10 $^{(*)}$ | 23.7 | 0.32 | **−1.02** | 0.34 $^{(*)}$ |
| | Ka-HS | 26.9 | 0.37 | −7.56 | 0.31 $^{(*)}$ | 29.2 | 0.47 | −11.6 | 0.09 $^{(*)}$ | 22.3 | 0.30 | −1.43 | 0.36 $^{(*)}$ |
| | Ka-MS | 24.7 | 0.34 | 4.62 | 0.30 $^{(*)}$ | 25.2 | 0.37 | −3.98 | 0.11 $^{(*)}$ | 25.4 | 0.33 | 7.17 | 0.34 |
| | Ku-NS | 26.8 | 0.35 | 11.0 | 0.44 | 27.5 | 0.36 | 9.84 | 0.42 | **19.6** | **0.25** | 6.99 | **0.70** |
| **10log$_{10}$(N$_w$)** (N$_w$ in mm$^{-1}$ m$^{-3}$) | DPR-NS | 14.0 | 4.63 | −2.95 | 0.17 $^{(*)}$ | **13.7** | **4.51** | −2.30 | 0.42 | 15.9 | 5.25 | −3.61 | 0.08 $^{(*)}$ |
| | DPR-MS | 14.4 | 4.63 | **1.00** | 0.19 $^{(*)}$ | 14.1 | 4.57 | −0.52 | **0.46** | 17.8 | 5.78 | **0.60** | −0.03 $^{(*)}$ |
| | DPR-HS | **13.1** | **4.14** | −3.08 | **0.35** $^{(*)}$ | 15.2 | 4.68 | 0.86 | −0.10 $^{(*)}$ | 13.8 | **4.36** | −4.21 | −0.07 $^{(*)}$ |
| | Ka-HS | 13.2 | 4.18 | −2.97 | 0.30 $^{(*)}$ | 15.1 | 4.65 | 0.94 | −0.07 $^{(*)}$ | 14.1 | 4.51 | **−5.35** | −0.09 $^{(*)}$ |
| | Ka-MS | 14.7 | 4.75 | −3.67 | 0.12 $^{(*)}$ | 16.2 | 5.14 | 0.10 | 0.07 $^{(*)}$ | 13.6 | 4.43 | −4.02 | 0.11 $^{(*)}$ |
| | Ku-NS | 14.2 | 4.69 | −4.23 | 0.14 $^{(*)}$ | 15.2 | 4.98 | −3.20 | 0.21 $^{(*)}$ | **13.4** | 4.43 | −4.69 | **0.22** $^{(*)}$ |

Figure 4a compares the rainfall rates obtained from the different GPM algorithms at the CFB with the disdrometer estimates at the ground. A dispersion of the data along the 1:1 line is evident, in particular for the higher precipitation. Looking at the merit factors reported in Table 5, the differences between the mean and point comparison modes are limited, with a slightly better performance of the latter in the NMAE. The optimal comparison mode obtains lower NMAE, MAE, and NB, and higher corr, suggesting that this comparison method yields more significant results than the other two. Overall, the DF and SF rain rate algorithms perform similarly, although comparing the DF and SF algorithms in terms of HS and MS modes, a slight improvement in the merit parameters is obtained when using the dual frequency information. DPR-HS is the one that performs the best in terms of NMAE, MAE, and corr. Considering all the cases, most of the correlation coefficients were within 0.6 and 0.8, with NMAE ranging from 30.3% to 73.4%. Considering the point comparison, the DPR-MS algorithm performs the best in terms of NB and corr, while the corresponding Ka-MS algorithm presents a very bad performance in terms of corr and MAE, and a worse NB with respect to DPR-MS. It is important to underline that

the maximum rain rate available for comparing satellite and ground-based data is around 10 mm h$^{-1}$, with most of the precipitation rate lower than 2 mm h$^{-1}$.

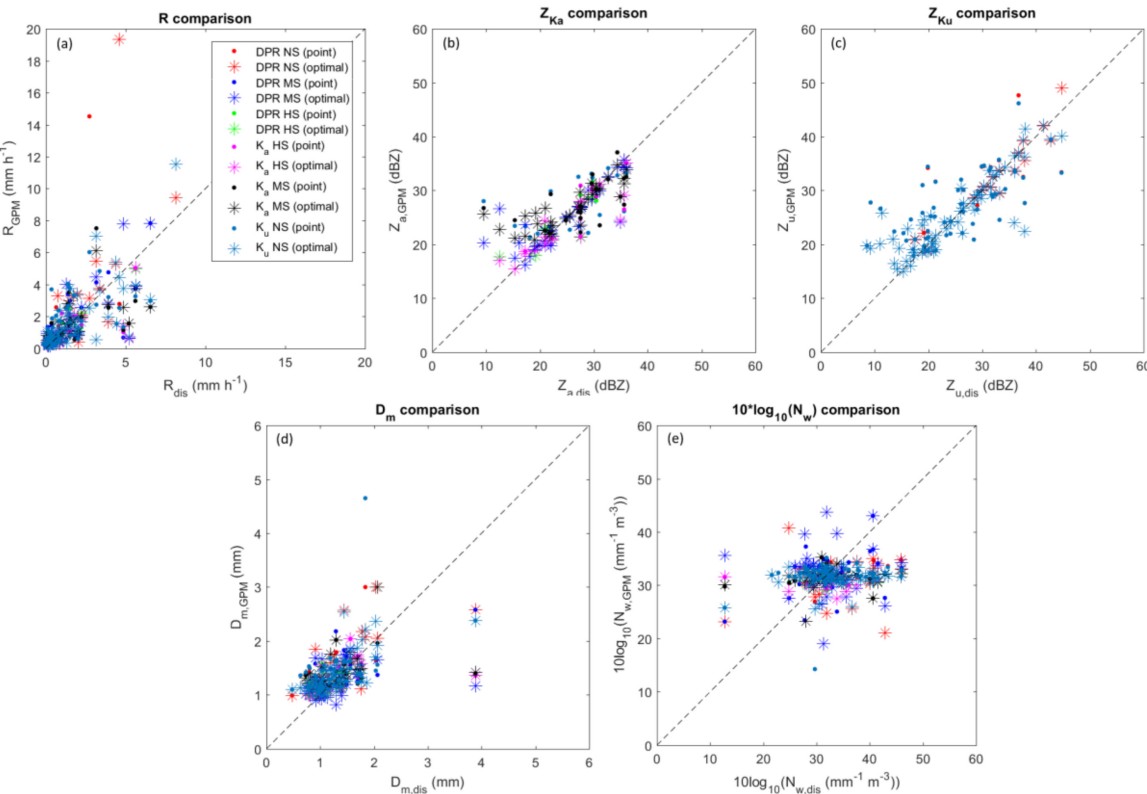

**Figure 4.** Scatterplot between rainfall rate (**a**), reflectivity factor (**b**) and (**c**), D$_m$ (**d**), and 10log$_{10}$(Nw) (**e**) obtained from disdrometers (x-axis) and the GPM products (y-axis) listed in the legend.

The scatterplot between the radar reflectivity factors obtained from GPM and disdrometers data is shown in Figure 4b,c for the Ka- and Ku-bands, respectively. A very small bias for both the variables (less than 10%, see Table 5) and a higher dispersion of the Ku-band data with respect to the Ka-band data is evident and results in higher NMAE and MAE values for the Ku-band (namely, the DPR-NS and Ku-NS modes) comparisons in Table 5. In terms of NMAE and correlation coefficients, the agreement between reflectivities (in decibel scale) is better than the one between rainfall rates. However, in terms of MAE, the values range between 2 dBZ and 5 dBZ. In this case, the difference in terms of mean and point comparison mode is limited, in particular, for the correlation coefficient. Again, the optimal comparison mode shows the best performance with correlation coefficients between 0.8 and 0.9, NMAE around 10% or less and NB less than 6%. However, it should be reminded that this mode optimizes the comparison with respect to the reflectivity. Furthermore, the improvement in terms of corrected reflectivity factor between DF and SF is small and affects mainly the MS and HS modes.

Figure 4d shows the scatterplot between satellite-based D$_m$ and the corresponding ground-based ones. The agreement is quite good, with NMAE less than 29%, for all the GPM products and for all the comparison modes, and NB less than 12%. However, the values of correlation coefficients are quite low. Small D$_m$ are overestimated, while D$_m$ greater than 1.5 mm are underestimated by GPM algorithms with respect to the disdrometer measurements. The results are compliant with the GPM science's requirements that require the mean D$_m$ to be within ±0.5 mm (see MAE values in Table 5). For DPR-MS, according to point and mean criteria, the performance is fairly good, with a NMAE of 22%, MAE lower than 0.3 mm and negligible bias, although such figures decrease in the optimal comparison mode. For some GPM products using the Ka-band measurements (i.e., DPR-

HS, Ka-HS, Ka-MS), the correlation coefficients obtained with the point comparison mode are unsatisfactory and considerably lower than those obtained for the optimal comparison mode. The performance of the Ku-NS mode appears, instead, better in the optimal mode.

The agreement in terms of $10\log_{10}N_w$ (with $N_w$ in the units of $\mathrm{mm^{-1}\ m^{-3}}$) is not satisfactory at all (Figure 4e). A similar conclusion was achieved in [45], although based on an earlier version of the GPM algorithms. Furthermore, our MAE results agree with results in [16]. In that work, the correlation coefficients were higher, but they used the GPM algorithm to retrieve $N_w$ from ground-based radar data, while in our study, the ground-based $N_w$ values are independent from the GPM retrieval since they are obtained from disdrometer data using Equation (7). In [45], differences between DPR (ver. 4) estimation of $N_w$ and the one obtained by a slightly different retrieval method that does not apply the DPR look-up table mechanism were also found. This may support that discrepancies between satellite-based and disdrometer–based $N_w$ can be due to the parameterization used by GPM to model the natural DSD. However, it is not easy to charge the lack of performance on $N_w$ in complex algorithms, such as those implemented for DPR, to a single cause. Finally, all the GPM retrieval algorithms are still subject to improvements and performances in $N_w$ retrieval could be improved by algorithm developers.

The differences between the mean, optimal and point comparison modes reported in Table 5 can be linked with the variability of precipitation in the pixels around the disdrometer. For this reason, Table 6 shows the standard deviation (std) of the considered variables within the 9 DPR pixels around the disdrometer (i.e., spatial variability). Furthermore, for completeness, also the standard deviation of the variable considering the disdrometer samples available within ±5 min from the GPM overpass time (i.e., temporal variability) is reported. In terms of rainfall rate, reflectivity, $D_m$ and $N_w$, the spatial and temporal variability are comparable.

**Table 6.** Standard deviation of the variables considering the 9 DPR pixels around the disdrometer and the disdrometer minutes around the GPM overpasses (labeled as disd).

| | | GPM (9 pixels around Disd.) | Disd (±5 min around GPM OverPasses) | | | GPM (9 Pixels around Disd.) | Disd (±5 min Around GPM OverPasses) |
|---|---|---|---|---|---|---|---|
| | | \multicolumn{6}{c}{**Std**} | | | |
| **R** (mm h$^{-1}$) | DPR-NS | 1.36 | 0.81 | **D$_m$** (mm) | DPR-NS | 0.24 | 0.20 |
| | DPR-MS | 0.90 | 0.82 | | DPR-MS | 0.24 | 0.20 |
| | DPR-HS | 0.52 | 0.79 | | DPR-HS | 0.18 | 0.25 |
| | Ka-HS | 0.53 | 0.97 | | Ka-HS | 0.20 | 0.25 |
| | Ka-MS | 0.69 | 0.89 | | Ka-MS | 0.18 | 0.21 |
| | Ku-NS | 1.13 | 0.81 | | Ku-NS | 0.21 | 0.20 |
| **Z** (dBZ) | DPR-NS | 4.13 | 4.01 | $10\log_{10}(N_w)$ ($N_w$ in mm$^{-1}$ m$^{-3}$) | DPR-NS | 2.44 | 2.38 |
| | DPR-MS | 3.25 | 3.53 | | DPR-MS | 3.30 | 2.48 |
| | DPR-HS | 3.30 | 3.76 | | DPR-HS | 0.99 | 2.44 |
| | Ka-HS | 3.24 | 3.90 | | Ka-HS | 1.20 | 2.46 |
| | Ka-MS | 2.41 | 3.47 | | Ka-MS | 1.31 | 2.45 |
| | Ku-NS | 4.14 | 4.01 | | Ku-NS | 1.04 | 2.38 |

It should be noted that the GPM values have a larger spread than that shown in Figure 4e in terms of $N_w$ since the data represented in this figure are limited to the values obtained in coincidence with disdrometer data or in the GPM pixel with the reflectivity value closest to the disdrometer one. In this regard, Figure 5 shows the $D_m$ vs. $10\log_{10}(N_w)$ obtained from GPM data during the GPM overpasses considering all the 9 pixels around the disdrometers (blue dots) and only the pixel over the disdrometer (red dots). In Figure 5a, all the SF GPM products are considered, while in Figure 5b, there are the DF products. Furthermore, as background reference, the disdrometers data have been plotted (grey dots). GPM data follow the typical $D_m$-$10\log_{10}(N_w)$ behavior depicted by grey dots, although most of the $10\log_{10}(N_w)$ data are concentrated around 30. The latter is true for both data in the 9 pixels around the disdrometer (blue dots; mean value 31.19) and the GPM data coincident with the disdrometer (red dots, mean value 31.87). The GPM-based $D_m$ retrievals appear to saturate around 3 mm, highlighting an artifact of the MS mode that limits the Ka-band

retrieval of $D_m$ to 3 mm [16]. The latter behavior is more evident in Figure 5b because the DF algorithms use both the Ka- and Ku-band retrievals.

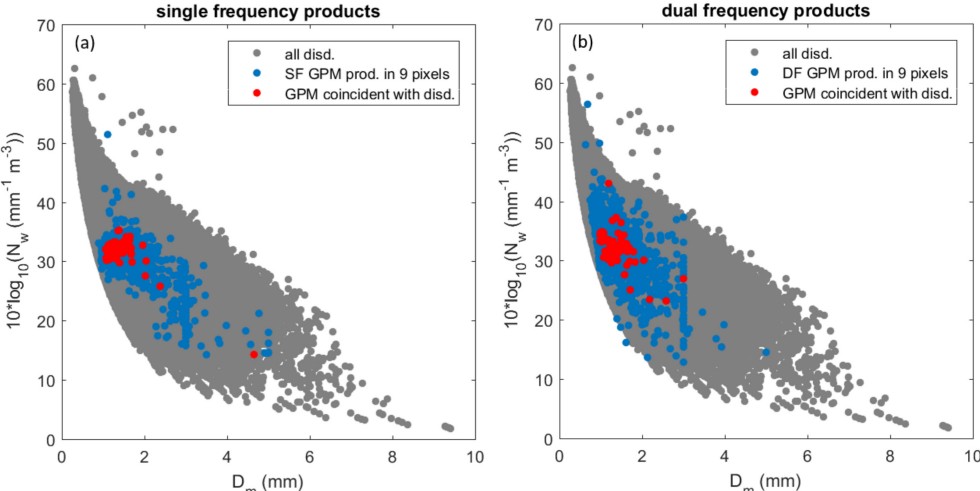

**Figure 5.** $D_m$ vs. $10\log_{10}(N_w)$ for (i) all the available disdrometer data (grey dots), (ii) the GPM data (all products) of the GPM pixel co-located with the disdrometer (red dots), and (iii) the GPM data (all products) in the 9 pixels around the disdrometers (blue dots). Left panel refers to SF products, while right panel to DF products.

Considering the precipitation type classification of DPR Level 2 files (namely the variable called TypePrecip), the comparison for R, Z, $D_m$, and $N_w$ was performed also considering stratiform and convective samples. This is important because the precipitation type determines the coefficients to be used in Equation (2). Classifying the precipitation types, the number of matched disdrometer and DPR data is considerably lower (i.e., see Table 7), also because for a certain number of DPR overpasses, the rainfall classification type is not reported in the GPM files, in particular, for the DF products. Therefore, only the results for the optimal comparison mode are reported (see Table 8). The results in Table 8 show that for stratiform rain, the agreement between disdrometer and satellite data is similar to the one obtained considering all the samples (see Table 5). For convective rainfall, it is difficult to provide some conclusions since the statistics are based on a very low number of samples. Furthermore, it should be noted that the results can be influenced by the averaging in time (namely over the 10-min disdrometer samples around the GPM overpasses time) that have different effects for stratiform and convective rain. In particular, the smoothing effect of the mean is more pronounced for convective rain than for stratiform one.

**Table 7.** Number of stratiform and convective samples obtained for the different GPM products considered in the optimal comparison mode.

|  | # of Sample | |
| --- | :---: | :---: |
|  | **Stratiform** | **Convective** |
| DPR-NS | 8 | 6 |
| DPR-MS | 3 | 10 |
| DPR-HS | 1 | 0 |
| Ka-HS | 20 | 0 |
| Ka-MS | 30 | 3 |
| Ku-NS | 60 | 6 |

**Table 8.** Merit parameters of the comparison between mean GPM and disdrometer data for stratiform and convective rainfall.

| | | Stratiform | | | | Convective | | | |
|---|---|---|---|---|---|---|---|---|---|
| | | NMAE (%) | MAE | NB (%) | Corr | NMAE (%) | MAE | NB (%) | Corr |
| **R** **(mm h$^{-1}$)** | DPR-NS | 50.4 | 1.05 | 11.2 | 0.54 | 40.0 | 0.31 | 29.4 | 0.76 |
| | DPR-MS | 18.4 | 0.18 | 16.1 | 0.66 | 39.3 | 0.57 | 8.46 | 0.76 |
| | DPR-HS | - | - | - | - | - | - | - | - |
| | Ka-HS | 42.9 | 0.62 | −31.8 | 0.66 | - | - | - | - |
| | Ka-MS | 49.2 | 0.75 | −6.69 | 0.69 | 63.7 | 1.34 | −51.1 | 0.64 |
| | Ku-NS | 53.3 | 0.77 | −0.59 | 0.74 | 33.7 | 0.82 | 13.0 | 0.67 |
| **Z** **(dB)** | DPRNS | 2.95 | 0.87 | −0.41 | 0.87 | 20.8 | 4.31 | 19.3 | 0.58 |
| | DPR-MS | 4.33 | 1.14 | −0.90 | 0.66 | 16.1 | 3.97 | 10.5 | 0.74 |
| | DPR-HS | - | - | - | - | - | - | - | - |
| | Ka-HS | 8.13 | 2.02 | −3.26 | 0.83 | - | - | - | - |
| | Ka-MS | 10.6 | 2.70 | 5.69 | 0.79 | 21.4 | 5.38 | 5.23 | 0.57 |
| | Ku-NS | 10.8 | 2.64 | 2.85 | 0.82 | 9.29 | 3.13 | −0.29 | 0.83 |
| **D$_m$** **(mm)** | DPR-NS | 11.8 | 0.16 | 8.87 | 0.79 | 26.3 | 0.29 | 13.8 | 0.38 |
| | DPR-MS | 12.4 | 0.17 | −7.29 | 0.40 | 25.7 | 0.32 | 10.4 | 0.56 |
| | DPR-HS | - | - | - | - | - | - | - | - |
| | Ka-HS | 22.3 | 0.30 | −1.43 | 0.36 | - | - | - | - |
| | Ka-MS | 25.3 | 0.34 | 5.66 | 0.34 | 25.8 | 0.28 | 25.8 | 0.30 |
| | Ku-NS | 18.7 | 0.23 | 8.81 | 0.60 | 27.5 | 0.54 | −4.15 | 0.61 |
| **10log$_{10}$(N$_w$)** **(N$_w$ in** **mm$^{-1}$ m$^{-3}$)** | DPR-NS | 12.5 | 4.27 | −7.04 | 0.12 | 10.9 | 3.53 | −0.40 | 0.28 |
| | DPR-MS | 9.66 | 2.96 | 5.13 | −0.67 | 15.0 | 4.99 | −3.24 | 0.38 |
| | DPR-HS | - | - | - | - | - | - | - | - |
| | Ka-HS | 14.1 | 4.51 | −5.35 | −0.09 | - | - | - | - |
| | Ka-MS | 13.4 | 4.30 | −2.70 | 0.12 | 15.5 | 5.71 | −15.5 | 0.08 |
| | Ku-NS | 12.7 | 4.26 | −5.11 | 0.02 | 24.9 | 7.14 | 0.31 | 0.30 |

Finally, as shown in Table 4, the number of matched overpasses is not the same for the different algorithms. Therefore, in order to better highlight the differences between DF and SF retrievals, in the following analysis, we considered the overpasses with (i) both the DPR-NS and Ku-NS products for the full swath and ii) the DPR-MS, DPR-HS, Ka-MS, and Ka-HS products for the inner swath. In this way, the results in terms of comparison with disdrometer data obtained for DPR-NS and Ku-NS are based on the same precipitation events. Similarly, the results for DPR-MS, DPR-HS, Ka-MS, and Ka-HS are obtained considering the same precipitation events. The results in terms of NMAE, MAE, NB, and corr obtained comparing the disdrometer data with the GPM retrievals of the above selected overpasses are shown in Figure 6. The colored bars represent the results for the optimal comparison mode, while the grey edge bars are for the point comparison mode. With a few exceptions, the agreement is better for the optimal comparison than for the point comparison, indicating that such approach is better for the validation of the satellite data. Comparing in Figure 6 the DF and SF algorithm for the same scan mode (namely NS, HS, or MS), we found that the performance of the DF algorithm is similar to the one of the SF. In most of the comparisons, the DF provides slightly better results (namely lower NMAE, MAE, and absolute NB and higher corr.); however, there are also cases where the opposite is true. Among MS and HS, the latter seems to perform a bit better, in particular, for *R* and *Z*.

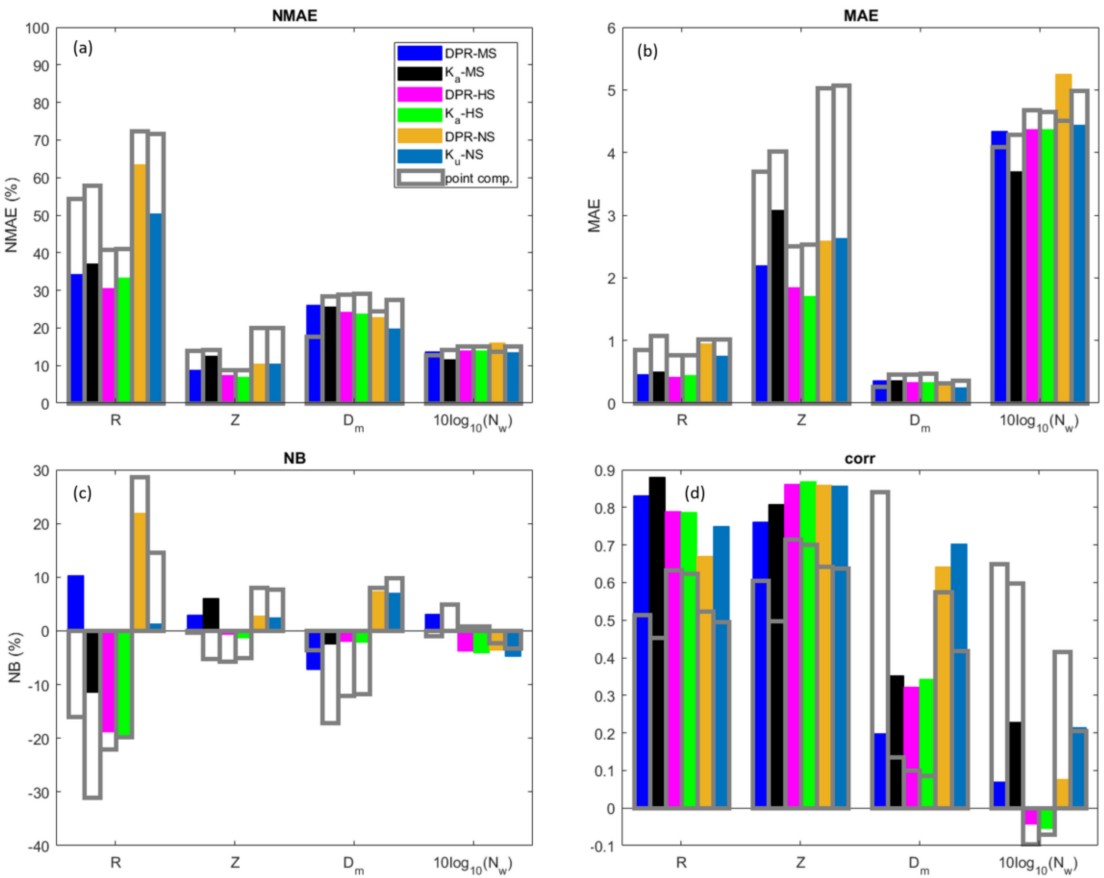

**Figure 6.** (**a**) NMAE, (**b**) MAE, (**c**) NB, and (**d**) corr obtained comparing disdrometer data with GPM overpasses with (i) DPR-NS and Ku-NS products available in the full swath and (ii) DPR-MS, DPR-HS, Ka-MS, Ka-HS available in the inner swath. The colored bars consider the optimal comparison, while the grey edge bars are for the point comparison.

## 6. Conclusions

The core satellite of the GPM mission was launched more than seven years ago, and since then, a big effort has been made to validate GPM products with ground-based devices. This study has presented the first validation of the GPM DPR observations in Italy with disdrometer data. To this purpose, data from seven different disdrometers located all over Italy were processed, analyzed, and compared with GPM DPR Level 2 products collected since the satellite launch date (2014) to October 2020. The comparison was performed considering the following variables: the reflectivity factors at the Ku and Ka bands corrected for attenuation, rainfall rate, and the DSD parameters $D_m$ and $N_w$. Depending on the DPR product, we found, in six years, a number of overpasses with precipitation over the considered disdrometer locations that ranges between 69 and 340. GPM DPR estimates are provided, for each footprint at ground, with a vertical nominal resolution of 125 m or 250 m. However, the lowest height at which estimates are performed corresponds to the lower range bin unaffected from clutter, although estimates are extrapolated to the range bin closest to the ground surface. A limited difference between the measurements obtained aloft and the ones estimated at ground level were observed, and therefore, we decided to use the data obtained at the first useful bin that can be, depending also on the disdrometer sites, between 0.60 km and 1.35 km above the disdrometer. Comparing point measurements at ground provided by the disdrometer with areal estimates aloft provided by GPM DPR requires specific strategies to take into account the different sampling of the two devices. Three different comparing modes are applied. The one that provides the best results is the optimal comparison mode that compares the disdrometer data with the DPR pixel with the closest reflectivity value in a $3 \times 3$ box around the pixel containing the disdrometer.

Considering all the four variables used for the comparison, a clear outperformance of the DF products is not evident, although, in most of the cases, the agreement is slightly better with respect to SF products. However, considering the comparison for rain rate and reflectivity, the DPR-HS product has the best agreement with disdrometer data. The latter results can be influenced by the fact that the DPR-HS data are provided only in the inner swath where measurements at Ka- and Ku-bands are available. Considering only the SF products, the Ka ones perform better for rain rate and reflectivity, while a better performance of Ku products is obtained for $D_m$ and $N_w$. Furthermore, it should be noted that mostly light-to-moderate precipitation intensities were available for the comparison and such a performance takes advantage from the sensitivity of the algorithms, which is indeed an important factor for precipitation measurements at mid- and high-latitudes. With higher rain rates, the performance of Ka products would be affected by attenuation effects. Therefore, a further analysis needs to be performed considering higher precipitation intensity regimes, but this requires also more disdrometers in order to increase the opportunities to sample intense precipitation. In general, the comparisons in terms of R, Z, and $D_m$ are satisfactory, while the comparison in terms of $10\log_{10}(N_w)$ is not. This suggests an in-depth investigation of the adopted retrieval algorithms to improve their performance.

**Author Contributions:** Conceptualization, E.A. and L.B.; software, E.A.; comparison analysis, E.A.; data curation and analysis, E.A., L.B., and M.M. for Rome site, F.P. and A.B. for Bologna site, V.C., C.A. and G.B. for Montevergine site; E.B. and A.L.Z. for Capua site, O.C. and G.C. for Milan site, R.B. and R.C. for Turin site and A.A. and A.O. for Florence site; writing—original draft preparation, E.A.; writing—review and editing, all the Authors, critical review, M.M., F.P. and L.B. All authors have read and agreed to the published version of the manuscript.

**Funding:** This research has been supported by the Institutions of the Authors. F.P. acknowledges EUMETSAT through the "Satellite Application Facility on Support to Operational Hydrology and Water Management" (H-SAF) for partially supporting this research. The V.C., and G.B. contribution to his work has been partially supported by the following project: R&I Project "Use of innovative technologies, materials and models in the aeronautic field (AEROMAT)", Line II "Support for Innovation",Specialization Area "Aerospace", call n. 1735/Ric. 13 July 2017—CUP code J66C18000490005, project identification code ARS01_01147, PON R&I 2014-2020. E.A., L.B., M.M. were partially supported by the European Space Agency under the activity "RainCast" (Contract: 4000125959/18/NL/NA).

**Data Availability Statement:** GPM DPR data can be downloaded from https://storm.pps.eosdis.nasa.gov/storm/ (accessed on 16 April 2021). Disdrometer data can be asked to the Authors.

**Acknowledgments:** The Authors acknowledge NASA and JAXA data processing teams for making the GPM data available. V.C., C.A. and G.B. are grateful to the Center for Monitoring and Modelling for Marine and Atmosphere applications of the University of Naples "Parthenope" (http://meteo.uniparthenope.it/) for providing disdrometric measurements collected in the Montevergine Observatory. E.A., M.M. and L.B. acknowledge ARPA Piemonte for having provided the Thies Clima disdrometer and the agreement between NASA and CNR ISAC for cooperation in the Hydrological Cycle in Mediterranean Experiment (HyMeX).

**Conflicts of Interest:** The authors declare no conflict of interest. The funders had no role in the design of the study; in the collection, analyses, or interpretation of data; in the writing of the manuscript, or in the decision to publish the results.

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
