# Peer review of "Validation of GPM Rainfall and Drop Size Distribution Products through Disdrometers in Italy"

_remotesensing, doi:10.3390/rs13112081_

Round 1
Reviewer 1 Report
The review on a manuscript "Validation of GPM rainfall and drop size distribution products through disdrometers in Italy" by Adirosi et al.
This paper describes validation of GPM satellite rainfall products against ground-based disdrometers in Italy. The authors compared rain rate, radar reflectivity factors, mass weighted mean diameter (Dm) and gamma DSD intercept parameter (Nw) retrievals obtained from different observational modes of the DPR (i.e. NS, MS and HS scan patterns for DPR, Ku and Ka measurements). They employed TC and P2 disdrometers installed in 7 ground-based observation sites across Italy. The paper is overall well-structured and well-written. The validation methodologies are examined thoroughly, including extrapolation effect in DPR surface retrievals and definitions of the matchup windows. Overall, I would recommend the paper for a minor revision in the present form.
Major Comments:
The authors defined an “Optimal” method where they selected a DPR pixel within 3x3 box around each disdrometer observation that gave the closest DPR radar reflectivity to the disdrometer estimate. This method does not sound scientifically adequate, and the results obtained are considered arbitrary. For example, if one of the 9 boxes exactly agrees the radar reflectivity but the rest of the 8 boxes have significant differences, the case is still evaluated as exact agreement. The combined evaluations of the “point” and “mean” methods that the authors already preformed are considered fairly sufficient for validating and addressing point versus areal measurement differences. If further consideration on precipitation spatiotemporal variability is required, why not derive standard deviation of the 3x3 box (as done in Table 6) and introduce a threshold where DPR and disdrometer estimates are compared only when the standard deviation is below the threshold and the precipitation is considered homogeneous enough to compare point versus areal measurements?
The presentation of Figures 4 and 6 could be improved. Different colours and symbols in Figure 4 are overlapped and it is difficult to understand and distinguish which of the DPR modes have better/worse correlations with the disdrometer measurements than others. I would recommend the authors to divide figures into groups by the DPR modes. In addition, the gray box in Figure 6 are presented in very thick lines and background colour bars are difficult to identify/compare.
How are the adequacy of the disdrometer evaluated and confirmed (before using them as “truth” for DPR validation)? Please explain in the manuscript.
The authors concluded that Nw gave a major discrepancies between DPR and disdrometer estimates. How are the Nw retrieval methods different between the DPR and disdrometer? Also, explain how the Nw estimates by disdrometers are “straightforwardly obtained” (line 707). Since this is one of the major findings of the work, please elaborate the descriptions in more detail to improve the scientific quality of the discussion.
Minor Comments:
Add a brief explanation on the physical meaning of the Dm and Nw.
Equation (1) – explain the parameter μ in the text.
Table 1 – Recommend adding “XXXX et al.” or “XXXX and YYYY” to each reference (in addition to the reference numbers).
Author Response
Please see attached file-

Reviewer 2 Report
It is a very well-presented draft, all seems fine with it.
Author Response
Authors thanks the Reviewer for the time spent in reviewing our manuscript.
Reviewer 3 Report
Generally this is a good and quite readable paper though lots of acronyms can make it hard going at times, I understand that really this is needed. I have a few comments, most concerning to me are points 4 and 5, without addressing these it is hard to judge the real performance of the GOM algorithms:
- Introduction could be improved, the gamma distribution is mentioned, but then only 2 parameters are derived, though the gamma function has a 3rd missing parameter, mu - the paper states it assumes mu=3, but gives no idea of the validity, or affects of this assumption. This section has no citations that should certainly be included.
- Section 2 of the paper is occasionally repetitive of the introduction, perhaps this can be avoided?
- Lines around 596 - you have done a sensitivity study into the averaging time used, but appear to have then selected the extreme longest you tested. Why not test longer averaging periods? At what period does the performance degrade?
- I feel it is important to consider the statistics in tables 5 and 6... are the results statistically significantly different from a climatology? Many look unlikely to be in my eyes. Perhaps bold and underlining can, rather than showing "best" and "worst" , show levels of p-value? This is most relevant to figure 6 where the paper says itself the sample sizes are very small. Without this information, its hard to judge the skill offered by GPM - especially given figure 5 suggests perhaps there is reduced specificity.
- Figure 5 is interesting and revealing - GPM data has much lower range in both parameters than the disdromters. Are the different disdrometers consistent? Why is the GPM such a reduced range? Probably averaging?
